# Partial eNOS Deficiency Results in Greater Levels of Vascular Inflammation and BBB Disruption in Response to Chronic Mild Hypoxia

**DOI:** 10.3390/ijms26167902

**Published:** 2025-08-15

**Authors:** Arjun Sapkota, Sebok K. Halder, Saifudeen Ismael, Gregory J. Bix, Richard Milner

**Affiliations:** 1San Diego Biomedical Research Institute, 3525 John Hopkins Court, Suite 200, San Diego, CA 92121, USA; asapkota@sdbri.org (A.S.); shalder@sdbri.org (S.K.H.); 2Department of Neurosurgery, Clinical Neuroscience Research Center, School of Medicine, Tulane University, New Orleans, LA 70112, USA; sismael@tulane.edu (S.I.); gbix@tulane.edu (G.J.B.); 3Department of Neurology, School of Medicine, Tulane University, New Orleans, LA 70112, USA; 4Department of Microbiology and Immunology, School of Medicine, Tulane University, New Orleans, LA 70112, USA; 5School of Public Health and Tropical Medicine, Tulane University, New Orleans, LA 70122, USA

**Keywords:** brain, aging, blood vessels, microglia, endothelial nitric oxide synthase (eNOS), chronic mild hypoxia, blood–brain barrier (BBB) integrity

## Abstract

Blood–brain barrier (BBB) deterioration with increasing age is an important factor contributing to vascular dementia. Previous studies show that endothelial nitric oxide synthase (eNOS) facilitates vascular endothelial growth factor-mediated angiogenesis and increased vascular permeability. In contrast, recent work has shown that aged hemi-deficient hemizygous eNOS^+/−^ mice manifest BBB disruption in association with increased incidence of thromboembolic events in the brain. To unravel whether eNOS contributes to or protects against hypoxia-induced cerebrovascular damage, we compared chronic mild hypoxia (CMH)-induced cerebrovascular angiogenic remodeling and BBB breakdown in aged (20 months old) eNOS^+/−^ and wild-type (WT) mice. This revealed that CMH strongly enhanced eNOS expression in cerebral blood vessels with much lower levels in eNOS^+/−^ mice. eNOS hemi-deficiency resulted in greater CMH-induced BBB disruption, but unexpectedly, had no effect on endothelial proliferation. eNOS^+/−^ mice also displayed enhanced endothelial expression of the endothelial activation markers MECA-32, VCAM-1, and β3 integrin in cerebral blood vessels, indicating greater vascular inflammation, and this correlated with increased levels of microglial activation and demyelination. Taken together, our results support the concept that eNOS plays an important protective function in the aged brain by suppressing endothelial activation and maintaining cerebrovascular health.

## 1. Introduction

Blood vessels in the central nervous system (CNS) are unique from those in other organs in having properties of high electrical resistance and low permeability. This unique feature of CNS blood vessels to carefully regulate what enters the CNS parenchyma to protect sensitive neural cells from potentially harmful blood components is referred to as the blood–brain barrier (BBB) [1,2,3,4,5]. At the structural level, the BBB consists of endothelial cells tightly attached to the underlying vascular basement membrane comprising extracellular matrix (ECM) proteins, the tight apposition of adjacent endothelial cells maintained by inter-endothelial tight junction proteins, and the influence of other CNS resident cell types including, pericytes, microglia, and astrocyte end-feet [6,7,8,9,10,11]. It is well established that BBB disruption is a critical pathogenic event in many different neurological conditions, including ischemic stroke, multiple sclerosis (MS), vascular dementia, and meningitis [12,13,14,15,16]. Accumulating evidence demonstrates that BBB integrity also declines with age [17,18].

Endothelial nitric oxide synthase (eNOS) generates nitric oxide (NO), which plays a fundamental protective role in regulating many aspects of vascular function. These include the promotion of vasodilation as well as facilitating vascular endothelial growth factor (VEGF)-mediated angiogenesis and increased vascular permeability [19]. Previous work using heterozygous eNOS^+/−^ mice showed that partial eNOS deficiency induces cerebral hypoperfusion and marked pathological changes in cerebral blood vessels that include thrombotic occlusions, particularly in the cerebral cortex and hippocampus, progressive cerebral amyloid angiopathy (CAA) and BBB breakdown [20]. Importantly, these changes correlated with cognitive impairment, prompting the authors to propose that eNOS^+/−^ mice represent an attractive model of vascular dementia, otherwise known as vascular contributions to cognitive impairment and dementia (VCID). A subsequent study demonstrated that aged eNOS^+/−^ mice have marked areas of demyelination, particularly in the corpus callosum and the hippocampus [21]. These pathological changes correlated closely with functional deficits in gait and associative recognition memory. Based on these findings, the authors proposed that partial eNOS deficiency leads to cerebral hypoperfusion and thrombus formation, which then precipitates BBB disruption, astrogliosis and subsequent demyelination and cognitive decline.

Our studies over the last 15 years have shown that chronic mild hypoxia (CMH; 8% O_2_) induces a robust vascular remodeling response in the CNS, culminating in 50% increased vessel density over a period of 2–3 weeks that is associated with transient BBB disruption [22,23]. These remodeling events are associated with an enhanced state of vascular activation, as shown by increased endothelial expression of several markers. The first is vascular cell adhesion molecule (VCAM)-1 whose primary role is to facilitate extravasation of peripheral leukocytes into the CNS. The second is mouse endothelial cell antigen (MECA)-32, a marker that is expressed by endothelial cells in the embryonic brain but then switched off around embryonic day E16 as the BBB matures, which is therefore absent in the normal postnatal brain but is reinduced on remodeling or inflamed cerebral endothelial cells in adult mice [24,25].

Based on our finding that aged (20 months) mice show much stronger hypoxia-induced BBB disruption (5–10-fold number of vascular leaks) compared to young (8–10 weeks), we hypothesized a link between hypoxic exposure and the pathogenesis of VCID [24]. In this model, we propose that the higher incidence of hypoxia in older people due to chronic lung and heart disease, together with the weaker BBB at this age, triggers BBB disruption, which then increases the risk of neuronal dysfunction, neurodegeneration and ultimately cognitive decline. As the CMH model displays marked and measurable changes in angiogenesis, BBB integrity, and vascular and glial cell activation [22,26], the aim of this study was to define the role of eNOS in regulating cerebrovascular remodeling and BBB disruption in the CMH model by comparing these events in aged (20 months old) eNOS^+/−^ and WT mice. We chose to study aged mice specifically because this is the age when the pathological events become most apparent both in the eNOS^+/−^ and CMH models, and because this is the age that is most translationally relevant as VCID occurs in the aging human population. Specific questions include the following: (i) do reduced eNOS levels stall or attenuate the hypoxia-induced vascular remodeling response, (ii) does lower eNOS activity also prevent or reduce CMH-induced BBB breakdown, and (iii) what effect does reduced eNOS activity have on glial cell reactivity and myelin integrity?

## 2. Results

### 2.1. Chronic Mild Hypoxia Enhances eNOS Expression

To determine how chronic mild hypoxia (CMH, 8% O_2_) influences cerebrovascular eNOS expression in aged (20 months old) wild-type (WT) and eNOS^+/−^ heterozygous mice, immunofluorescence (IF) analysis of brain sections was performed (Figure 1A). This revealed that 4 days of CMH induced a significant increase in eNOS expression in WT mice (*p* < 0.01) and eNOS^+/−^ heterozygous mice (*p* < 0.01) that was associated with an apparent widening of cerebral blood vessels, consistent with our previous observations [27]. This analysis also confirmed that eNOS expression is significantly reduced in eNOS^+/−^ heterozygous mice both under normoxic and hypoxic conditions (*p* < 0.001) as shown in Figure 1A,B. The specificity of the eNOS antibody is demonstrated in Figure 1C, which shows total absence of eNOS staining in eNOS^−/−^ (KO) mice (lower row) as well as strong colocalization of the eNOS signal with the endothelial specific marker CD31 (upper row).

### 2.2. eNOS^+/−^ Mice Show Greater Hypoxia-Induced BBB Disruption, but Endothelial Proliferation Is Not Affected

Previous studies have shown that eNOS plays an important role regulating several aspects of vascular function, including promoting angiogenic remodeling and vascular permeability [19,28]. Because CMH triggers a marked angiogenic response that is associated with transient BBB disruption, we set out to define the impact of eNOS in influencing these activities by examining CMH-induced cerebral angiogenesis and BBB disruption in eNOS^+/−^ heterozygous mice. Based on previous findings of other labs [19,28], we anticipated that when exposed to CMH, eNOS^+/−^ mice would show attenuated levels of BBB disruption and angiogenesis compared to wild-type (WT) mice. To examine this, we exposed WT and eNOS^+/−^ mice to 4 days of CMH and then performed IF analysis of frozen brain tissue. BBB disruption was evaluated using CD31 to label blood vessels and fibrinogen to denote extravascular leak following BBB disruption (Figure 2A,B). Consistent with previous results [24], while the brains of normoxic mice (both WT and eNOS^+/−^ strains) showed no extravascular fibrinogen leak, those exposed to hypoxia displayed numerous extravascular fibrinogen deposits. Contrary to our expectation of seeing fewer leaks in eNOS^+/−^ mice, we observed a significantly higher density of vascular leaks compared to WT mice (*p* < 0.05 in the midbrain and a consistent trend in the olfactory bulb; Figure 2A,B). Additional studies using extravascular hemoglobin leak as an alternative marker of BBB disruption confirmed these findings (*p* < 0.05 in both the midbrain and olfactory bulb; Figure 2C,D). Next, to determine if reduced eNOS levels attenuate the CMH-induced angiogenic response, we examined endothelial proliferation, an important early stage of the angiogenic response, by performing CD31/Ki67 dual-IF in WT and eNOS^+/−^ brains. This revealed a total absence of proliferating brain endothelial cells in normoxia treated mice of both strains, but a strong increase in proliferating endothelial cells in both strains of mice under hypoxic conditions (Figure 3A). Notably, however, there was no appreciable difference in the brain endothelial proliferation rate of eNOS^+/−^ and WT mice (Figure 3B). As it is possible that the angiogenic rate could differ between the two mouse strains at other steps of the vascular remodeling process, we also quantified the final endpoint of angiogenesis, namely the total vascular area after 4 days of CMH. This showed a marked increase in vessel area after 4 days of CMH in both strains of mice. Interestingly, in contrast to our expectations, the hypoxia-induced increase in total vascular area was significantly higher in eNOS^+/−^ mice compared to WT (*p* < 0.001; Figure 3C). These results demonstrate that hemi-deficiency of eNOS results in greater hypoxia-induced BBB disruption, no difference in the endothelial proliferation rate but a small yet significant increase in total vascular area.

### 2.3. Cerebral Blood Vessels in eNOS^+/−^ Mice Are More Activated

As eNOS has been described as having a protective influence on vascular health [20,21], we next examined if reduced levels of eNOS could be producing a heightened state of vascular activation, and that this might be predisposing to increased risk of vascular breakdown. To examine this, we evaluated several endothelial markers that are upregulated during vascular activation/inflammation, including mouse endothelial cell antigen (MECA)-32, vascular cell adhesion molecule (VCAM)-1, and β3 integrin. MECA-32 is a marker that is expressed by endothelial cells in the embryonic brain but is then switched off around embryonic day E16 as the BBB matures. It is therefore absent in the postnatal brain but is reinduced on remodeling or inflamed cerebral endothelial cells in adult mice [24,25]. Figure 4A shows that MECA-32 is absent on both strains of mice under normoxic conditions but is induced following exposure to hypoxia. Importantly, the density of MECA-32^+^ vessels was significantly upregulated in the eNOS^+/−^ strain compared to the WT following hypoxia (Figure 4B). VCAM-1 expression is generally limited to medium size cerebral vessels under normoxic conditions (Figure 4C). Notably while hypoxia increased the density of VCAM-1^+^ vessels in the brains of both mouse strains, it was significantly higher in the eNOS^+/−^ strain vs. WT (Figure 4D). β3 integrin is another adhesion molecule that is absent on resting endothelium but is strongly induced on angiogenic cerebral endothelial cells under hypoxic conditions [29] (Figure 4E). Quantification revealed that β3 integrin expression was markedly increased on cerebral blood vessels of eNOS^+/−^ mice (Figure 4F). In summary, these studies demonstrate that compared to WT mice, cerebral endothelial cells in eNOS^+/−^ mice express higher levels of activation molecules under hypoxic conditions that is associated with an increased level of BBB disruption.

### 2.4. Enhanced Vascular Activation in eNOS^+/−^ Mice Correlates with Increased Levels of Microglial Activation and Demyelination

As cerebral blood vessels in eNOS^+/−^ mice are more activated/inflamed and show an increased susceptibility to disrupt under hypoxic conditions we next evaluated how this impacts the behavior of cells downstream within the brain parenchyma. First, we evaluated how reduced levels of eNOS impacts microglial activation by examining microglial Mac-1 expression and morphology. As shown in Figure 5, this revealed that under normoxic conditions, microglia in eNOS^+/−^ mice express significantly higher levels of Mac-1 which correlated with a more activated morphology (thicker cell body). Exposure to CMH triggered enhanced microglial activation in WT mice though had no significant impact in the eNOS^+/−^ mice. These findings suggest that while microglia in the normoxic WT brain occupy a lower activation state which can then be further enhanced by hypoxia. In contrast, microglia in the eNOS^+/−^ brain are almost fully activated under normoxic conditions and have limited ability to show further activation by the hypoxia stimulus. Next, in light of the fundamental role for myelin in ensuring higher cerebral functioning, we examined the impact of eNOS hemi-deficiency on myelin integrity within the major forebrain myelinated tract, the corpus callosum. Under normoxic conditions fluoromyelin IF revealed high levels of myelin integrity in WT mice, but some patchy significant levels of erosion were observed in eNOS^+/−^ mice (Figure 5C,D). Consistent with our recently published data, WT mice exposed to CMH displayed marked loss of myelin integrity [30], but notably, this hypoxia-induced demyelination was amplified in eNOS^+/−^ mice.

## 3. Discussion

Previous studies have demonstrated that eNOS regulates vascular function by facilitating vascular endothelial growth factor (VEGF)-mediated angiogenesis and increased vascular permeability [19]. This led us to speculate that reduced levels of eNOS in eNOS^+/−^ hemi-deficient mice might attenuate both the angiogenic and BBB disruption events triggered by CMH. In contrast to these expectations, other studies have shown that aged eNOS^+/−^ mice manifest increased BBB disruption in association with increased incidence of thromboembolic events in the brain [20,21]. To unravel this apparent contradiction, here we compared hypoxia-induced cerebrovascular angiogenic remodeling and BBB breakdown in aged WT mice and mice hemi-deficient in eNOS (eNOS^+/−)^. Our main findings were as follows: (i) CMH strongly enhanced eNOS expression in cerebral blood vessels and eNOS^+/−^ mice displayed markedly suppressed eNOS levels, (ii) eNOS hemi-deficiency resulted in greater CMH-induced BBB disruption, but unexpectedly, had no effect on endothelial proliferation, (iii) eNOS reduction also resulted in enhanced endothelial expression of the endothelial activation markers MECA-32, VCAM-1, and β3 integrin, indicating increased vascular inflammation, and (iv) this correlated with increased levels of microglial activation and greater levels of demyelination in eNOS^+/−^ mice. Taken together, our results support the concept that eNOS plays important protective functions in maintaining cerebrovascular health. This is borne out by our finding that even 50% reduction in cerebral eNOS levels results in significantly greater levels of vascular activation and BBB disruption correlating with amplified microglial activation and myelin damage.

### 3.1. Hypoxic Enhancement of Cerebrovascular eNOS Expression

In this study, we found that CMH strongly induced eNOS expression in cerebral blood vessels. This is consistent with other studies showing that chronic hypoxia increases eNOS expression in vascular endothelial cells of the adult lung in vivo as well as in vitro [31,32,33]. In contrast, other groups found that prolonged hypoxia downregulated eNOS levels in the rat aorta [34] while another study of fetal blood vessels in rabbits showed that chronic intermittent hypoxia increased eNOS levels in carotid but had the opposite effect in femoral arteries [35]. These conflicting results suggest that the impact of hypoxia on eNOS level is likely context dependent and may be influenced by several factors, including the duration of hypoxia, the organ in question, age, and the disease model being examined.

### 3.2. Role of eNOS in Angiogenesis

One surprising result from our study was that rates of hypoxia-induced endothelial proliferation and increased vascularity were not attenuated in eNOS^+/−^ mice; indeed, we observed a small but significant increased degree of CNS vascularity in the eNOS^+/−^ strain. This was unexpected because overwhelming evidence supports a central role for eNOS in driving many aspects of angiogenic remodeling, including endothelial proliferation, survival, and migration [19,28,36,37], so we predicted that 50% reduction in eNOS levels would result in diminished angiogenic responses. One explanation that might account for this is that because the hypoxic-angiogenic response in the brain is so much more extensive than in other (peripheral) organs, it is possible that other mechanisms may compensate in the brain. These include the hypoxia-inducible factor (HIF)-1α-VEGF axis, as well as other pro-angiogenic mechanisms including the angiopoietin 2-Tie2 axis [38,39,40] and the fibronectin-α5β1 integrin axis [16]. The alternative explanation is that because eNOS is so important, 50% eNOS levels may be sufficient to maintain an optimal angiogenic response. That we observed a small but highly significant increase in total vascularity following hypoxic exposure in eNOS^+/−^ mice could also suggest that a compensatory mechanism may have been activated to offset the partial loss of eNOS. Further analysis in future experiments should shed light on which of these possibilities is most likely.

### 3.3. The Influence of eNOS on Vascular Activation and BBB Permeability

One of the most striking and robust findings from these studies was that hemi-deficiency of eNOS resulted in enhanced levels of hypoxic-induced vascular activation, as seen by the marked upregulation of the endothelial activation markers MECA-32, VCAM-1, and β3 integrin, which correlated with increased levels of BBB disruption. Based on a prior landmark study describing an important role for eNOS in mediating VEGF-induced angiogenesis and vascular permeability [19], we predicted that partial loss of eNOS would result in partial protection from hypoxia-induced BBB disruption, but in fact we observed the opposite result. What could account for this? One possibility is that the chronic nature of prolonged eNOS reduction has a cumulative effect in eNOS^+/−^ mice. In keeping with the findings of Fukumura et al. [19], it is possible that an acute reduction in eNOS in an otherwise normal healthy mouse may indeed lead to reduced cerebrovascular permeability. However, chronic reduction of eNOS in eNOS^+/−^ mice has been shown to lead to a range of pathological sequelae including cerebral hypoperfusion, thromboembolism, glial activation and BBB disruption [20,21]. Based on these observations it seems likely that the brains of these mice have an elevated pro-inflammatory environment that would lead to our observed findings of elevated vascular activation and increased BBB disruption. An alternative view of this is that the effects of eNOS on cerebral blood vessels are age-dependent and that while lower eNOS may reduce BBB permeability in young healthy mice, in the more pro-inflammatory aged brain, opposite effects are seen. In future studies we plan to study these effects in young mice to see if this is the case.

Growing evidence demonstrates that BBB integrity is reduced with age [17,18,41,42]. Taken with the increased risk of hypoxic episodes in older people as a result of chronic lung disease (asthma, pulmonary fibrosis, emphysema), heart disease (ischemic heart disease, heart failure), sleep apnea, and increased risk and severity of life-threatening chest infections (pneumonia) [43,44,45,46,47], it becomes clear that the combination of these two age-related events greatly increases the risk of hypoxia-induced BBB disruption. This increases the risk of neuronal dysfunction, neurodegeneration and ultimately will culminate in cognitive decline. Our current findings demonstrate that eNOS is an important molecular component that suppresses endothelial activation, reduces BBB disruption, and maintains cerebrovascular health, findings that are highly translationally relevant to a growing elderly human population.

## 4. Materials and Methods

### 4.1. Animals

The studies described were reviewed and approved by the Institutional Animal Care and Use Committee at San Diego Biomedical Research Institute (SDBRI). Homozygous eNOS^−/−^ mice were obtained from Jackson Laboratories and bred with C57BL6/J mice maintained under pathogen-free conditions in the closed breeding colony of SDBRI to produce heterozygous eNOS^+/−^ and wild-type (WT) mice, which were genotyped and selected for experiments. 

### 4.2. Chronic Hypoxia Model

Aged (20 months old) eNOS^+/−^ and WT female mice or young adult (4 months old) female eNOS^−/−^ (KO) and WT mice were housed 5 to a cage, and placed into a hypoxic chamber (Biospherix, Redfield, NY, USA) maintained at 8% O_2_ for 4 days. Littermate control mice were kept in the same room under similar conditions except that they were kept at ambient sea-level oxygen levels (normoxia, approximately 21% O_2_ at sea level) for the duration of the experiment. Every few days, the chamber was briefly opened for cage cleaning and food and water replacement as needed.

### 4.3. Immunohistochemistry and Antibodies

Immunohistochemistry was performed on 10 µm frozen sections of cold phosphate-buffered saline (PBS) perfused tissues as described previously [27]. Rat monoclonal antibodies from BD Biosciences reactive for the following antigens were used in this study: CD31 (clone MEC13.3; 1:300), Mac-1 (clone M/170; 1:50), MECA-32 (1:100) and VCAM-1 (1:100). The hamster anti-β3 integrin monoclonal antibody (clone 2C9.G2; 1:100) was also from BD Biosciences. Rabbit antibodies reactive for the following proteins were also used: Ki67 (1:4000 from Novus Biologicals, Centennial, CO), fibrinogen (1:2000 from Millipore, Temecula, CA), hemoglobin (1:20,000 from Cloud Clone Corp, Katy, TX, USA), and eNOS (1:1500 from ThermoFisher Scientific, Carlsbad, CA, USA). Fluoromyelin red (1:50) was obtained from Invitrogen (Carlsbad, CA, USA). Secondary antibodies used included Cy3-conjugated anti-rabbit (1:1000) and anti-rat (1:500) from Jackson Immunoresearch (West Grove, PA, USA), and Alexa Fluor 488-conjugated anti-rat (1:500) from Invitrogen.

### 4.4. Image Analysis

Images were taken using a 5×, 10× or 20× objective on an Axioskop2 plus microscope (Carl Zeiss, Dublin, CA, USA) equipped with an Infinity 3S camera (Lumenera, Ottawa, ON, Canada) and Infinity Analyze imaging software (Lumenera version 6.5.5). For each antigen, images of at least three randomly selected areas were taken at 5×, 10× or 20× magnification per tissue section and three sections per brain analyzed to calculate the mean for each animal (n = 4 mice per group). For each antigen in each experiment, exposure time was set to convey the maximum amount of information without saturating the image and was maintained constant for each antigen across the different experimental groups. Most analyses in this study (Figure 1, Figure 2, Figure 3, Figure 4 and Figure 5A,B) were performed in the midbrain, specifically in the ventral tegmental area (VTA), while analysis for Figure 5C,D was performed in the corpus callosum. The number of fibrinogen^+^ or hemoglobin^+^ vascular leaks or MECA-32^+^, VCAM-1^+^ or β3 integrin^+^ vessels per field of view (FOV) was quantified by capturing images and performing manual counts. The % eNOS^+^, Mac-1^+^, CD31^+^, or fluoromyelin^+^ area was measured and quantified by measuring the % area/FOV using Image J software version 1.53t. Endothelial proliferation was quantified by counting the number of CD31/Ki67 dual-positive cells per FOV. Each experiment was performed with 4 different animals per condition, and the results expressed as the mean ± SEM. Statistical significance was assessed using one-way analysis of variance (ANOVA) followed by Tukey’s multiple comparison post hoc test or Student’s t test, in which *p* < 0.05 was defined as statistically significant. All images (normoxia and hypoxia) within a single panel of a figure are presented at the same magnification.

## 5. Conclusions

In this study, we examined whether eNOS contributes to or protects against hypoxia-induced cerebrovascular damage by comparing CMH-induced cerebrovascular angiogenic remodeling and BBB breakdown in aged (20 months old) eNOS^+/−^ and wild-type (WT) mice. We first determined that CMH strongly enhanced eNOS expression in cerebral blood vessels and that eNOS^+/−^ mice displayed much lower levels. eNOS hemi-deficiency resulted in greater CMH-induced BBB disruption as revealed by multiple markers, but unexpectedly, had no effect on endothelial proliferation. Of note, cerebral blood vessels in eNOS^+/−^ mice displayed enhanced expression of the endothelial activation markers MECA-32, VCAM-1, and β3 integrin, indicating greater vascular inflammation, and this correlated with increased levels of microglial activation and demyelination. Together, our results support the concept that eNOS plays an important protective function in the aged brain by suppressing endothelial activation and maintaining cerebrovascular health.

## Figures and Tables

**Figure 1 ijms-26-07902-f001:**
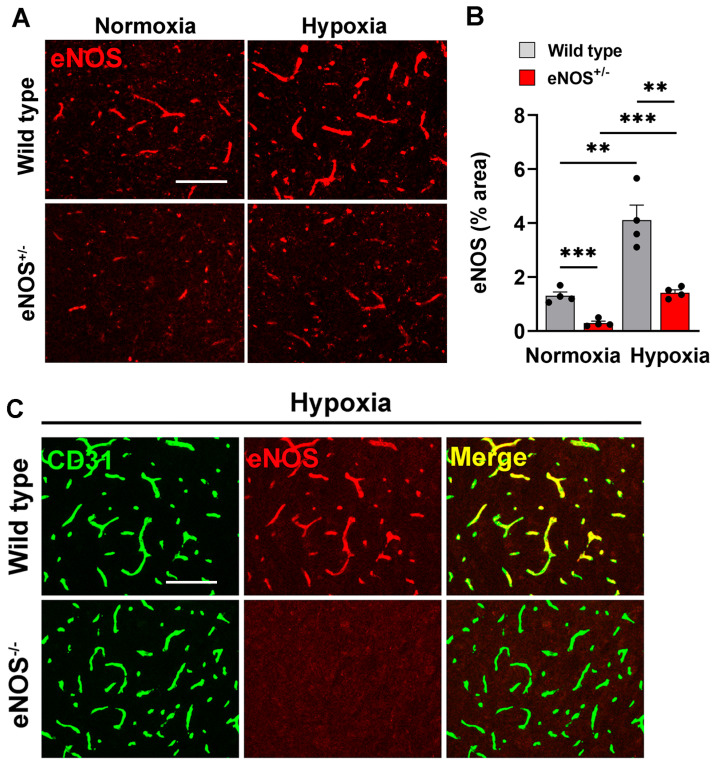
Hypoxic induction of eNOS in cerebral blood vessels. (**A**) Frozen brain sections taken from aged (20 months old) eNOS^+/−^ or WT mice exposed to normoxia or 4 days of hypoxia (8% O_2_) were immunostained for eNOS. Images were captured in the midbrain. Scale bar = 100 μm. (**B**) Quantification of eNOS expression in the midbrain after normoxia or 4 days of hypoxia. All results are expressed as the mean ± SEM (n = 4 mice/group). ** *p* < 0.01, *** *p* < 0.001. Note that hypoxia increases eNOS expression in cerebral blood vessels and that eNOS^+/−^ mice express lower levels compared to WT both under normoxic and hypoxic conditions. (**C**) Frozen brain sections taken from adult (4 months old) eNOS^−/−^ (KO) or WT mice exposed to 4 days hypoxia (8% O_2_) were dual-immunostained for CD31 (AlexaFluor-488) and eNOS (Cy-3). Note the strong colocalization of the eNOS signal with the endothelial specific marker CD31 (upper row) as well as the absence of any eNOS signal in eNOS^−/−^ KO mice.

**Figure 2 ijms-26-07902-f002:**
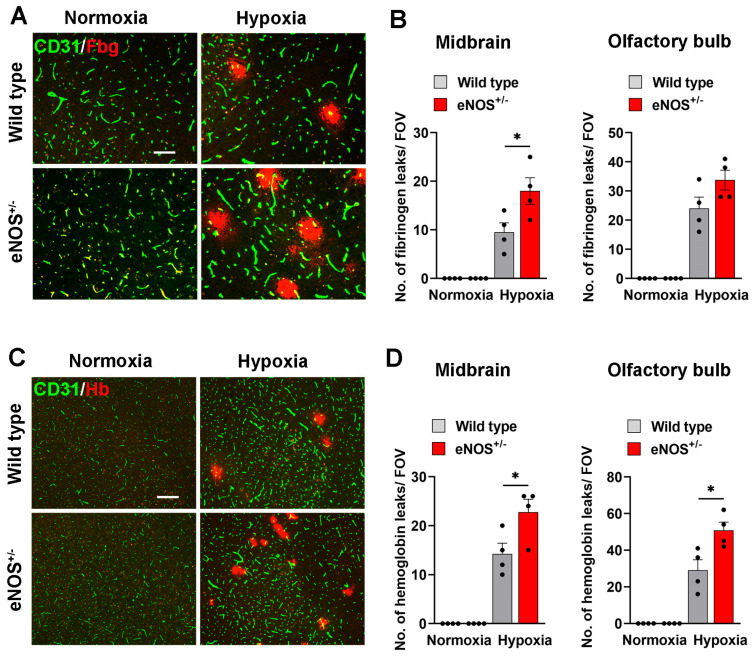
eNOS^+/−^ mice show greater levels of hypoxia-induced BBB breakdown. Frozen brain sections taken from aged (20 months old) eNOS^+/−^ or WT mice exposed to normoxia or 4 days hypoxia (8% O_2_) were dual-stained for CD31 (AlexaFluor-488) and fibrinogen (Fbg) (Cy-3) (**A**) or CD31 (AlexaFluor-488) and hemoglobin (Hb) (Cy-3) (**C**). Images show the midbrain region. Scale bar = 100 μm (**A**) or 200 μm (**C**). Quantification of the number of fibrinogen^+^ leaks/FOV (**B**) or hemoglobin^+^ leaks/FOV (**D**) in the midbrain and olfactory bulb after normoxia and 4 days of hypoxia. Results are expressed as the mean ± SEM (n = 4 mice/group). * *p* < 0.05. Note that eNOS^+/−^ mice displayed greater levels of hypoxia-induced BBB breakdown compared to WT mice.

**Figure 3 ijms-26-07902-f003:**
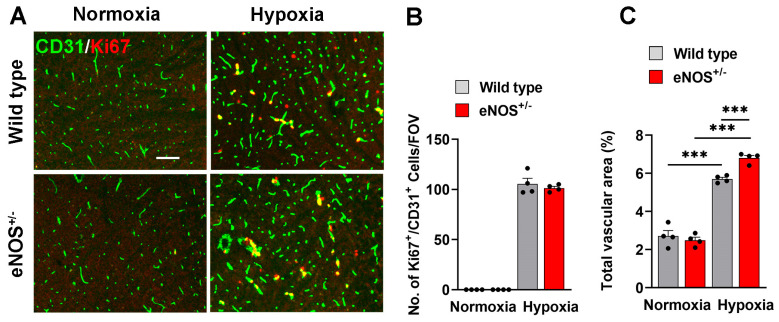
Analysis of angiogenic remodeling in eNOS^+/−^. (**A**) Frozen brain sections taken from aged (20 months old) eNOS^+/−^ or WT mice exposed to normoxia or 4 days hypoxia (8% O_2_) were dual-stained for CD31 (AlexaFluor-488) and Ki67 (Cy-3). Images show the midbrain region. Scale bar = 100 μm. (**B**) Quantification of the number of proliferating brain endothelial cells (**B**) or vessel density (**C**) after normoxia or 4 days hypoxia. Results are expressed as the mean ± SEM (n = 4 mice/group). *** *p* < 0.001. Note that eNOS^+/−^ mice showed no difference in endothelial proliferation rate compared to WT mice.

**Figure 4 ijms-26-07902-f004:**
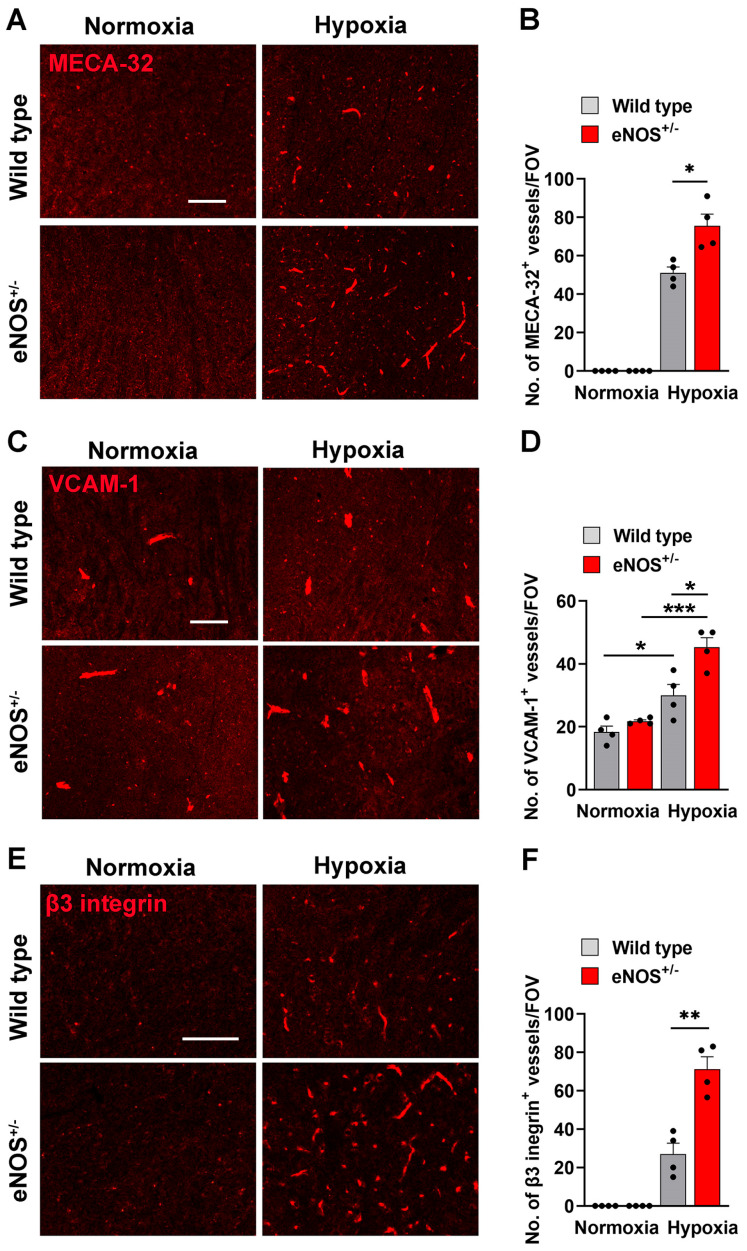
Cerebral blood vessels in eNOS^+/−^ mice show higher levels of activation markers. Frozen brain sections taken from aged (20 months old) eNOS^+/−^ or WT mice exposed to normoxia or 4 days hypoxia (8% O_2_) were immunostained for MECA-32 (**A**), VCAM-1 (**C**) or b3 integrin (**E**)**.** Images show the midbrain region. Scale bar = 100 μm. Quantification of the number of MECA-32^+^ (**B**), VCAM-1^+^ (**D**) or β3 integrin^+^ (**F**) vessels/FOV in the midbrain after normoxia or 4 days hypoxia. Results are expressed as the mean ± SEM (n = 4 mice/group). * *p* < 0.05, ** *p* < 0.01, *** *p* < 0.001. Note that under normoxic conditions, expression levels of the three activation markers were similar between the two strains of mice. However, under hypoxic conditions, eNOS^+/−^ mice displayed significantly greater levels of all three activation markers.

**Figure 5 ijms-26-07902-f005:**
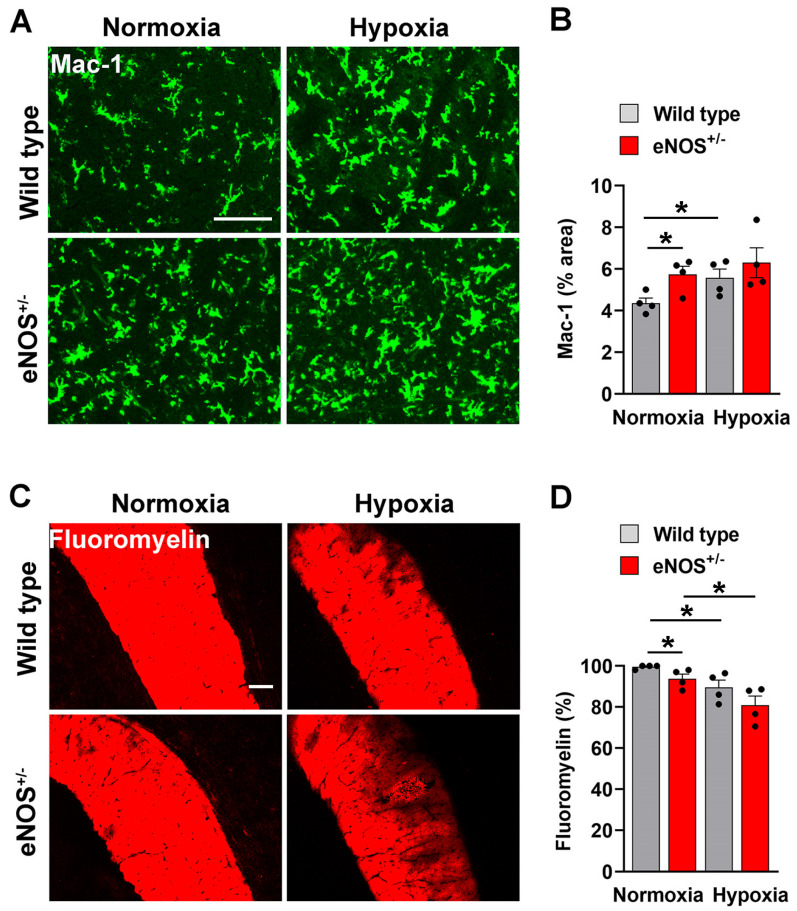
eNOS^+/−^ mice show enhanced levels of microglial activation and demyelination. Frozen brain sections taken from aged (20 months old) eNOS^+/−^ or WT mice exposed to normoxia or 4 days hypoxia (8% O_2_) were stained for Mac-1 in the midbrain (**A**) or fluoromyelin in the corpus callosum (**C**). Scale bar = 100 μm. Quantification of the Mac-1^+^ (**B**) or fluoromyelin^+^ (**D**) area after normoxia or 4 days hypoxia. Results are expressed as the mean ± SEM (n = 4 mice/group). * *p* < 0.05. Note that Mac-1 expression was greater in eNOS^+/−^ mice under normoxic conditions, and that hypoxia enhanced Mac-1 expression in WT mice but had only marginal effect in eNOS^+/−^ mice. Interestingly, demyelination was observed in eNOS^+/−^ mice even under normoxic conditions, and hypoxia triggered loss of myelin both in WT and eNOS^+/−^ strains.

## Data Availability

The datasets used and/or analyzed during the current study are available from the corresponding author upon reasonable request.

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
