# Peer review of "Partial eNOS Deficiency Results in Greater Levels of Vascular Inflammation and BBB Disruption in Response to Chronic Mild Hypoxia"

_ijms, 2025, doi:10.3390/ijms26167902_

Round 1
Reviewer 1 Report
Comments and Suggestions for Authors
The manuscript primarily investigates the changes in eNOS expression under hypoxic conditions, as well as the alterations in Blood-brain barrier (BBB) epithelial cell proliferation and the endothelial activation markers MECA-32, VCAM-1, and β3 integrin in cerebral blood vessels upon eNOS deficiency. Through a multi-dimensional approach, it confirms the important protective role of eNOS in maintaining cerebrovascular health in the aging brain. The study presents a certain degree of novelty and provides a basis for further research on the role of eNOS in age-related blood-brain barrier changes. However, the study has the following issues:
- It is recommended to supplement the Introduction section with a discussion on the roles of blood-brain barrier-related activation markers such as MECA-32 and VCAM-1 in BBB disruption, as detected in the experiments.
- Why were only female mice used for the hypoxia animal experiments instead of including both sexes equally? Additionally, why was the sample size limited to only four mice?
- In Figures 5A and 5B, there is no significant difference in the Mac-1 positive staining area between hypoxic eNOS+/- mice and normoxic wild-type mice. Please provide an explanation for this result.
- The current experiments infer differences in blood-brain barrier disruption indirectly by detecting staining patterns in BBB vascular epithelial cells. Could experiments be designed to directly assess vascular permeability changes in hypoxic eNOS+/- model mice by measuring the levels of substances crossing the blood-brain barrier?
- It is suggested that scale bars be added to all images, rather than only the first one.
Author Response
Reviewer #1:
- “It is recommended to supplement the Introduction section with a discussion on the roles of blood-brain barrier-related activation markers such as MECA-32 and VCAM-1 in BBB disruption, as detected in the experiments.”
We have included some description of these vascular activation markers within the Introduction section as requested by the Reviewer:
These remodeling events are associated with an enhanced state of vascular activation, as shown by increased endothelial expression of several markers. The first is vascular cell adhesion molecule (VCAM)-1 whose primary role is to facilitate extravasation of peripheral leukocytes into the CNS. The second is mouse endothelial cell antigen (MECA)-32 a marker that is expressed by endothelial cells in the embryonic brain but then switched off around embryonic day E16 as the BBB matures, which is therefore absent in the normal postnatal brain but is reinduced on remodeling or inflamed cerebral endothelial cells in adult mice [23, 24].
- “Why were only female mice used for the hypoxia animal experiments instead of including both sexes equally? Additionally, why was the sample size limited to only four mice?”
We used female mice only in these studies primarily because most of our previous work describing hypoxia-induced vascular remodeling events has been performed in female mice and thus we have a clear idea of the baseline events in wild-type female mice. We are currently comparing hypoxic responses of female and male mice and plan to describe these findings more completely in a subsequent publication. The number of mice was limited to 4 for these studies primarily because of economic reasons.
- “In Figures 5A and 5B, there is no significant difference in the Mac-1 positive staining area between hypoxic eNOS+/- mice and normoxic wild-type mice. Please provide an explanation for this result.”
To be clear, we did not directly compare levels of Mac-1 expression between hypoxic eNOS+/- mice and normoxic wild-type mice because this is not a very meaningful comparison for this study. The key observations in this figure are that under normoxic conditions, microglia eNOS +/- mice are more activated than WT mice, and that WT mice show increased microglial activation when switched from normoxia to hypoxia. Interestingly, Exposure while CMH triggered enhanced microglial activation in WT mice it had no significant impact in the eNOS+/- mice. We believe the most likely reason for this is that while microglia in the normoxic WT brain occupy a lower activation state, this can then be further enhanced by hypoxia. By contrast, microglia in the eNOS+/- brain are already very activated under normoxic conditions and have limited ability to show further activation by the hypoxia stimulus.
- “The current experiments infer differences in blood-brain barrier disruption indirectly by detecting staining patterns in BBB vascular epithelial cells. Could experiments be designed to directly assess vascular permeability changes in hypoxic eNOS+/- model mice by measuring the levels of substances crossing the blood-brain barrier?”
We agree with the Reviewer that showing changes in vascular permeability is an important readout in these studies. In fact, Figure 2 depicts vascular permeability changes in eNOS +/- mice by evaluating extravascular leak of the endogenous blood proteins fibrinogen and hemoglobin. In future studies we plan to build on this by evaluating vascular leak of exogenous tracers such as FITC-dextrans of different molecular sizes.
- “It is suggested that scale bars be added to all images, rather than only the first one.”
We totally agree with the Reviewer that it is imperative that scale bar information be provided in images. To our knowledge it is a well-established standard protocol to include a scale bar within just one image of a set of any group of images within each figure as we have done in all figures in this manuscript. This has always been our modus operandi and is the standard acceptable approach.
Reviewer 2 Report
Comments and Suggestions for Authors
Authors compared chronic mild hypoxia (CMH)-induced cerebrovascular angiogenic remodelling and BBB breakdown in aged eNOS+/- and wild type mice. These results sup-port the concept that eNOS plays an important protective function in the aged brain by suppressing endothelial activation and maintaining cerebrovascular health.
Manuscript is interesting, corresponds to the journal theme, but the paper has some problems. The writing of the paper should meet the Journal standard.
Some comments:
-What about novelty of the proposed method and approach? Please give information about novelty of this study by comparing with your and similar studies (Halder SK, Milner R. Exaggerated hypoxic vascular breakdown in aged brain due to reduced microglial vasculo-protection. Aging Cell. 2022 Nov;21(11):e13720. doi: 10.1111/acel.13720; Halder SK, Milner R. The impact of chronic mild hypoxia on cerebrovascular remodelling; uncoupling of angiogenesis and vascular breakdown. Fluids Barriers CNS. 2021 Nov 17;18(1):50. doi: 10.1186/s12987-021-00284-x, https://doi.org/10.1186/s12987-023-00453-0).
-Figs 1 and 2 in the article https://doi.org/10.1186/s12987-023-00453-0 and Figs. 1 and 2 in the present manuscript are very similar. Please explain the novelty.
-Why “The aim of this study was to define the role of eNOS in regulating cerebrovascular remodelling and BBB disruption in the CMH model” wrote in discussion section? It is should be in Introduction section.
-If the “The aim of this study was to define the role of eNOS in regulating cerebrovascular remodelling and BBB disruption in the CMH model” – Please answer on it question and let the possible solutions to this problem.
-Authors wrote “These conflicting results suggest that the impact of hypoxia on eNOS level is likely context dependent and may be influenced by a number of factors, including the duration of hypoxia, the organ in question, age, and the disease model being examined.” - This conclusion is obvious even without these experiments.
-Discussion section is very poor with obvious conclusions in advance. Please add more information.
-Authors should make their manuscript and experiment results stand out in some way.
-Please highlight research importance.
-Conclusion section is lost. Please add it.
-There is no scale bar in Figs. 1-5 A. Add it.
Author Response
Reviewer #2:
- “What about novelty of the proposed method and approach? Please give information about novelty of this study by comparing with your and similar studies (Halder SK, Milner R. Exaggerated hypoxic vascular breakdown in aged brain due to reduced microglial vasculo-protection. Aging Cell. 2022 Nov;21(11):e13720. doi: 10.1111/acel.13720; Halder SK, Milner R. The impact of chronic mild hypoxia on cerebrovascular remodelling; uncoupling of angiogenesis and vascular breakdown. Fluids Barriers CNS. 2021 Nov 17;18(1):50. doi: 10.1186/s12987-021-00284-x, https://doi.org/10.1186/s12987-023-00453-0).”
The novelty of the current manuscript is that here we have specifically examined how hemi-deficiency of eNOS (in eNOS +/- mice) impacts the hypoxia-induced vascular remodeling events that we have described in some of our previous publications including those mentioned by the Reviewer.
- “Figs 1 and 2 in the article https://doi.org/10.1186/s12987-023-00453-0 and Figs. 1 and 2 in the present manuscript are very similar. Please explain the novelty.”
It is true that at first glance the first two figures in the current paper and that cited by the Reviewer may seem similar, but a closer examination reveals that they are very different. In the article cited by the Reviewer, Figure 1 displays hypoxic induction of fibronectin and a5 integrin on blood vessels in both young and aged brain. In contrast, Figure of the current manuscript under review shows how hypoxia upregulates eNOS expression and also how eNOS levels are markedly lower in eNOS +/- mice. Figure 2 in the cited manuscript shows the impact of blocking b1 integrin function on hypoxia-induced BBB disruption (using extravascular fibrinogen leak) in both young and aged mice whilst Figure 2 in the manuscript under review shows how hypoxia-induced BBB disruption (using extravascular leak of both fibrinogen and hemoglobin) is greater in eNOS +/- mice. So to be clear, the figures 1 and 2 in these different papers provide very different information.
- “Why “The aim of this study was to define the role of eNOS in regulating cerebrovascular remodelling and BBB disruption in the CMH model” wrote in discussion section? It is should be in Introduction section.”
We have taken the Reviewer’s advice and removed this sentence from the start of the Discussion section and incorporated it into the last paragraph of the Introduction section as follows:
As the CMH model displays marked and measurable changes in angiogenesis, BBB integrity, and vascular and glial cell activation [18, 21], the aim of this study was to define the role of eNOS in regulating cerebrovascular remodelling and BBB disruption in the CMH model by comparing these events in aged (20 months old) eNOS+/- and WT mice.
- “If the “The aim of this study was to define the role of eNOS in regulating cerebrovascular remodelling and BBB disruption in the CMH model” – Please answer on it question and let the possible solutions to this problem.”
As described in the Discussion section, the clear goal of these studies was to ascertain whether partial loss of eNOS in aged mice would protect or worsen hypoxia-induced BBB disruption. This is important because on the one hand, some studies have demonstrated that eNOS promotes angiogenesis and increased vascular permeability, but on the other hand other studies have shown that aged eNOS+/- mice manifest increased BBB disruption in association with increased incidence of thromboembolic events in the brain [16, 17]. The answer was very clear, namely that partial eNOS loss resulted in greater BBB disruption and enhanced vascular activation. Together this provides a very clear message that at least in the aged brain, eNOS plays a very beneficial function by suppressing vascular activation and disruption.
- “Authors wrote “These conflicting results suggest that the impact of hypoxia on eNOS level is likely context dependent and may be influenced by a number of factors, including the duration of hypoxia, the organ in question, age, and the disease model being examined.” - This conclusion is obvious even without these experiments.”
To be fair, the sentence highlighted by the Reviewer was used to summarize the outcomes of a number of different studies in which the effect of hypoxia on eNOS in different organs, age of the animal, and duration of hypoxia was different. The bottom line is that the role played by eNOS appears to be different depending on these several factors. This is why in the Discussion section, we point out that in future studies we aim to repeat these studies in young mice to see if outcomes in young mice differ from aged mice.
- “Discussion section is very poor with obvious conclusions in advance. Please add more information.”
I’m not sure what the Reviewer means by this. Certainly, before we performed these studies it was not at all clear what the outcomes would be. Based on prior studies, one could predict either that reduced levels of eNOS might lead to protection from hypoxia-induced BBB disruption and less endothelial proliferation, or conversely, that it would result in worsening of BBB disruption. As it turned out, our studies yielded clear unequivocal answers that have high translational significance as pertaining to the growing elderly human population with increased incidence of hypoxia-causing conditions such as lung and heart disease.
To address this criticism, we added another paragraph to the Discussion section as follows:
Growing evidence demonstrates that BBB integrity is reduced with age [13, 14, 39, 40]. Taken with the increased risk of hypoxic episodes in the aged as a result of chronic lung disease (asthma, pulmonary fibrosis, emphysema), heart disease (ischemic heart disease, heart failure), sleep apnea, and increased risk and severity of life-threatening chest infections (pneumonia) [41-45], it becomes clear that the combination of these two age-related events greatly increases the risk of hypoxia-induced BBB disruption. This increases the risk of neuronal dysfunction, neurodegeneration and ultimately will culminate in cognitive decline. Our current findings demonstrate that eNOS is an important molecular component that suppresses endothelial activation, reduces BBB disruption, and maintains cerebrovascular health, findings that are highly translationally relevant to a growing elderly human population.
- “Authors should make their manuscript and experiment results stand out in some way.”
We believe that the apparent contradiction set up by previous studies was begging to be addressed and in tis study, we took the novel approach to evaluating the overall effect of partial eNOS deficiency in the chronic mild hypoxia model, which provides clear readouts in many aspects of cerebrovascular function, including remodeling, angiogenesis, BBB disruption, endothelial activation, and downstream impact on other cell functions, including microglia and myelin-forming cells oligodendrocytes. In this respect, our manuscript stands out by clarifying what role eNOS plays in aged mice under hypoxic insult, a situation that is highly translationally relevant to a growing elderly human population.
- “Please highlight research importance.”
We have included the main take home message in the Discussion section as well as in the added “Conclusions” section to better address this point.
- “Conclusion section is lost. Please add it.”
We apologize for this oversight and have now inserted a Conclusion section in the revised manuscript.
- “There is no scale bar in Figs. 1-5 A. Add it.”
Please see the answer to Reviewer #1 point 5.
We absolutely agree with the Reviewer that it is imperative that scale bar information be provided in images. To our knowledge it is a well-established standard protocol to include a scale bar within just one image of a set of images within each figure as we have done in all figures in this manuscript. This has always been our modus operandi and is the standard acceptable approach. There simply isn’t the need to insert a scale bare into every panel of a multi-panel figure.
Reviewer 3 Report
Comments and Suggestions for Authors
Comments to Authros
MSS Ref: ijms-3739698
In this manuscript entitled “Partial eNOS deficiency results in greater levels of vascular inflammation and BBB disruption in response to chronic mild hypoxia” Sapkota. A et al., described that eNOS hemi-deficiency resulted in greater CMH-induced BBB disruption, but unexpectedly, had no effect on endothelial proliferation. eNOS+/- mice also displayed enhanced endothelial ex-pression of the endothelial activation and author claimed that that eNOS plays an important protective function in the aged brain by suppressing endothelial activation and maintaining cerebrovascular health. The experiment designed in 20-month-old mice is an excellent concept, however at this extended age multiple other things are also happening which might play crucial role despite the fact that authors used appropriate control.
To determine eNOS expression Chronic mild hypoxia enhances immunofluorescence immunohistochemistry was used but authors stated that “we used immunofluorescence (IF) to quantify eNOS-----” this looks like a strange statement.
Since this immunofluorescence localization is one of the major component of the present study and frozen brain sections which are prone to autofluorescence. In that case authors need to provide a control for specificity of staining.
Figure 1, also showed significant morphological changes in cerebral blood vessels but did not discuss.
This is indicating that images are captured from midbrain, better provide the exact brain regions.
Not sure but need a clarification whether magnification of Images from Normoxia and hypoxia is same.
Author Response
Reviewer #3:
- “To determine eNOS expression Chronic mild hypoxia enhances immunofluorescence immunohistochemistry was used but authors stated that “we used immunofluorescence (IF) to quantify eNOS-----” this looks like a strange statement.”
To make this clearer, we reworded the opening sentence in the Results section thus:
To determine how chronic mild hypoxia (CMH, 8% O2) influences cerebrovascular eNOS expression in aged (20 months old) wild type (WT) and eNOS+/- heterozygous mice, immunofluorescence (IF) analysis of brain sections was performed (Figure 1).
- “Since this immunofluorescence localization is one of the major component of the present study and frozen brain sections which are prone to autofluorescence. In that case authors need to provide a control for specificity of staining.”
We agree with the Reviewer that brain sections can sometimes come with a degree of autofluorescence. In our experience, this only becomes a problem if the intensity of the antibody staining is particularly weak, such as with a low abundance protein. However, this is not a problem with any of the stainings in this manuscript and certainly not with the eNOS staining shown in Figure 1. To address this criticism, we included an additional panel C in Figure 1 which shows total lack of any eNOS staining signal in the brains of eNOS -/- (KO) mice. This panel also highlights the close colocalization of eNOS signal with the endothelial cell marker CD31 in wild type mice, demonstrating that eNOS is expressed specifically within endothelial cells. We also included the following text in the Results section to reinforce this point.
The specificity of the eNOS antibody is demonstrated in Figure 1C, which shows total absence of eNOS staining in eNOS-/- (KO) mice (lower row) as well as strong colocalization of the eNOS signal with the endothelial specific marker CD31 (upper row).
- “Figure 1, also showed significant morphological changes in cerebral blood vessels but did not discuss.”
This is a good point noticed by the Reviewer and it is something we observe constantly when mice are exposed to CMH- an apparent widening as part of the remodeling of blood vessels. We have now included the following text in the Results section to point this out:
This revealed that 4 days CMH induced a significant increase in eNOS expression in WT mice (p < 0.01) and eNOS+/- heterozygous mice (p < 0.01) that was associated with an apparent widening of cerebral blood vessels, consistent with our previous observations [24].
- “This is indicating that images are captured from midbrain, better provide the exact brain regions.”
We have now included an extra sentence in the “Image analysis” sub-section of the Materials and Methods section to describe which specific brain areas the analyses were performed. Text was included to clarify this as below:
Most analyses in this study (Figures 1-4 and 5A) were performed in the midbrain, specifically in the ventral tegmental area (VTA), while analysis for Figure 5B was performed in the corpus callosum.
- ‘Not sure but need a clarification whether magnification of Images from Normoxia and hypoxia is same.”
We have included a sentence at the end of the “Image analysis” sub-section of the Materials and Methods section to confirm that all images within a single panel of a Figure are presented at the same magnification.
All images (normoxia and hypoxia) within a single panel of a Figure are presented at the same magnification.
Note: We have also reduced our self-citation rate to 20% (9 out of 45 total) to conform to the stipulations of the journal.
Round 2
Reviewer 2 Report
Comments and Suggestions for Authors
The Authors addressed all the questions and the manuscript is now improved.